# Contralesional Cortical and Network Features Associated with Preoperative Language Deficit in Glioma Patients

**DOI:** 10.3390/cancers14184469

**Published:** 2022-09-15

**Authors:** Chunyao Zhou, Shengyu Fang, Shimeng Weng, Zhong Zhang, Tao Jiang, Yinyan Wang, Lei Wang, Kai Tang

**Affiliations:** 1Department of Neurosurgery, Beijing Tiantan Hospital, Capital Medical University, Beijing 100050, China; 2Beijing Neurosurgical Institute, Capital Medical University, Beijing 100071, China; 3Research Unit of Accurate Diagnosis, Treatment and Translational Medicine of Brain Tumors, Chinese Academy of Medical Sciences, Beijing 100071, China

**Keywords:** glioma, language performance, subcortical network, graphic theory, voxel-based morphometry, diffusion tensor image

## Abstract

**Simple Summary:**

Gliomas that infiltrate eloquent areas can damage the corresponding cortical or subcortical structures, leading to language dysfunction. A total of 20–40% of eloquent area glioma patients have language deficits. Gliomas anchored in eloquent areas cause varying degrees of language impairment. A tumor’s size, grade, location, and contralesional compensation may explain these differences. This study aimed to retrospectively explore gray and white matter plasticity in the contralesional hemisphere of patients with gliomas in the eloquent area using VBM and DTI network analysis.

**Abstract:**

Lower-grade Gliomas anchored in eloquent areas cause varying degrees of language impairment. Except for a tumor’s features, contralesional compensation may explain these differences. Therefore, studying changes in the contralateral hemisphere can provide insights into the underlying mechanisms of language function compensation in patients with gliomas. This study included 60 patients with eloquent-area or near-eloquent-area gliomas. The participants were grouped according to the degree of language defect. T1 and diffusion tensor imaging were obtained. The contralesional cortical volume and the subcortical network were compared between groups. Patients with unimpaired language function showed elevated cortical volume in the midline areas of the frontal and temporal lobes. In subcortical networks, the group also had the highest global efficiency and shortest global path length. Ten nodes had intergroup differences in nodal efficiency, among which four nodes were in the motor area and four nodes were in the language area. Linear correlation was observed between the efficiency of the two nodes and the patient’s language function score. Functional compensation in the contralesional hemisphere may alleviate language deficits in patients with gliomas. Structural compensation mainly occurs in the contralesional midline area in the frontal and temporal lobes, and manifests as an increase in cortical volume and subcortical network efficiency.

## 1. Introduction

Lower-grade gliomas are slow-growing neoplasms of the central nervous system [1]. Gliomas that infiltrate eloquent areas can damage the corresponding cortical or subcortical structures, leading to language dysfunction [2,3]. A total of 20–40% of eloquent area glioma patients have language deficits [3,4], such as aphasia [5], anomia [4,6], or spontaneous speech defects [7]. The occurrence of language defects may be associated with tumor location [8], tumor grade [9], or the distance between the tumor border and the eloquent area (SDTN) [3,8]. However, in clinical practice, the severity of aphasia may vary among patients with a similar tumor location and size [5,9,10,11].

The compensation mechanism has been well described [12]. Activation of the compensation ability often alleviates original tumor-induced dysfunction [6,13]. Lesions or trauma can activate the brain’s structural and functional potential in its undamaged parts [14], manifested by increased cortical volume [15] and enhanced functional connectivity [16]. Previous studies have suggested that functional compensation mostly occurs in the area around the lesion [6,13,16], while more recent findings indicate that for lesions anchored in eloquent areas, the homotopic area on the opposite side of the lesion is more likely to be the area where functional compensation occurs [12,15,17,18,19,20]. Thus, we propose that the severity of language impairment in patients with eloquent-area gliomas may be associated with language function compensation in the contralesional hemisphere.

Multimodal magnetic resonance imaging (MRI) technology is a common method for studying brain function compensation [12]. Voxel-based morphometry (VBM) is a high-resolution T1 based voxel-level analysis method that can quantitatively measure differences in cortex volume between cohorts [21]. Meanwhile, diffusion tensor imaging (DTI) can measure subcortical structures and further explore white matter changes at the whole-brain level by constructing a white matter connection network and performing topological analysis. Quantitative network analysis combining high-resolution T1 with DTI is an ideal solution for observing both global and microstructural changes in a patient’s brain.

In this study, patients with gliomas invading or nearly invading the language areas in the dominant hemisphere were included and grouped according to naming and comprehensive language function. Herein, we aimed to explore gray and white matter plasticity in the contralesional hemisphere of glioma patients in the eloquent-area using VBM and DTI network analysis.

## 2. Materials and Methods

### 2.1. Patients

A total of 60 patients diagnosed with WHO grade II or III gliomas between September 2018 and December 2021 at Beijing Tiantan Hospital were retrospectively enrolled in this study. The inclusion criteria were as follows: (a) age above 18 years; (b) primary diagnosis of glioma; (c) tumors located in eloquent areas or near-eloquent areas in the left hemisphere; and (d) right-handedness. The eloquent area on each case was identified based on Broadmann anatomical atlas (BA 39,40,44,45). The exclusion criteria were as follows: (a) contraindications for MRI; (b) head motion > 1 mm in translation or 3° in rotation; (c) obvious bilateral glioma invasion observed on T1 or T2 images; and (d) history of antiepileptic drug use before MRI acquisition. Written consent was obtained from all patients, and the local review board approved the study.

### 2.2. MRI Acquisition

A MAGNETOM Prisma 3T MR scanner (Siemens, Erlangen, Germany) was used to perform conventional MRI with the following scanning parameters. Anatomic images were collected with T1 (TR = 2300 ms, TE = 2.3 ms, flip angle = 8 o, field of view (FOV) = 240 × 240 mm, voxel size = 1.0 × 1.0 × 1.0 mm^3^, slice number = 192) and T2 (TR = 3200 ms; TE = 87 ms; FA = 150°; FOV = 220 × 220 mm; voxel size = 0.9 × 0.9 mm × 5 mm; slice number = 25). DTI data were acquired using a single shot, echo planar imaging sequence (TR = 6000 ms, TE = 103 ms, axial slices = 75, resolution = 2.0 × 2.0 × 2.0 mm, flip angle = 75°; FOV = 230 × 230 mm; voxel size = 2.0 × 2.0 × 2.0 mm, number of directions = 30, b = 0/1000 s/mm^2^, EPI factor = 154).

### 2.3. Awake Craniotomy Protocol

All patients underwent awake craniotomy and language mapping with direct cortical bipolar stimulators (Ojemann stimulators; bipolar interval distance = 5 mm). This procedure was clarified in our previous study [8]. A Chinese language picture-naming task was applied (http://cgga.org.cn/language_test/index.jsp). If the stimulation spot caused speech arrest, anomia, dysarthria, or slowed speech, it was labeled as a potential positive site. If more than two continuous stimulations (2/3 principle) caused a positive reaction, the spot was defined as a positive site. Moreover, we used sterile circles with a 5 mm diameter to mark each positive site and acquired the central point of the circle using an intraoperative neuro-navigation system. For each glioma, parts of the tumor showing positive sites were saved, and the remaining tumor was removed. Drainage veins lying on the surface of the cortex were carefully preserved.

After the language area was set, the shortest length would be visually identified on the navigation system, the method was introduced in our former research [8]. Based on the coordinates of the language area (X_l_, Y_l_, Z_l_) and the point located on the border of the tumor (X_t_, Y_t_, Z_t_), the shortest distance was calculated with the formula as follow.
$$D=\sqrt{{\left(X_{l}-X_{t}\right)}^{2}+{\left(Y_{l}-Y_{t}\right)}^{2}+{\left(Z_{l}-Z_{t}\right)}^{2}}$$

### 2.4. Language Function Assessment

The western aphasia battery test was used to evaluate the preoperative language status of each patient within 24 h before tumor resection. Based on the test scores, the patients were first divided into anomia patients (AP) and non-anomia patients (nAP). The naming task is one of the main cognitive functions at risk of decline as language impairment occurred [22]. Thus, based on the calculation formula of the test score and the proportion of the score of each task [23], if the score of the naming task was lower than 9.8, the patients in the nAP group were classified as having mild naming defects and classified as mild anomia patients (mAP). 

### 2.5. Tumor Segmentation and Voxel-Based Morphometry

The tumor masks were manually segmented by two experienced neurosurgeons based on T2-FLAIR. Tumor volumes were calculated, while the masks were normalized and overlapped (Figure 1) by SPM. High-resolution T1 images were preprocessed using the VBM 8 based on statistical parametric mapping (https://www.fil.ion.ucl.ac.uk/spm/). Briefly, the process included normalization, segmentation, and smoothing. First, the images were normalized to the Montreal Neuroscience Institute space based on a linear transformation algorithm. All individual brains were then segmented into the cortex, white matter, and CSF. Subsequently, all segmented cortex images were smoothed using an 8-mm full-width at half-maximum Gaussian kernel. Visual checks of the processed images were implemented after every step to exclude apparent anatomical deviation. A hemispheric mask was used to segment all cortical volumes into left (ipisileisional) and right (contralesional) hemisphere, only contralesional cortical volume was transfer into further analysis. Finally, the preprocessed images were grouped and transferred to a general linear model (GLM) for further group-wise analysis.

### 2.6. DTI Preprocessing

Preprocessing and analysis of diffusion metrics were performed using the pipeline toolbox PANDA (http://www.nitrc.org/projects/panda/), which was based on the FMRIB software library (https://fsl.fmrib.ox.ac.uk/fsl/fslwiki/, accessed on 2 June 2019). The detailed pre-processing, tractography, and network construction steps were clarified by Cui et al. [24]. First, the program implemented steps to extract basic DTI metrics, including brain mask extraction, eddy current effect correction, averaging of multiple acquisitions, diffusion tensor calculation, and metric production. deterministic fiber tracking on contralesional hemisphere was implemented using the following standard: angle threshold = 45° and FA threshold = 0.2.

### 2.7. White Matter Connectome Construction

To construct the contralesional WM connectome matrix, deterministic fiber tracking was used to track white matter connections for all possible node pairs. We selected all brain regions in the right hemisphere in the “brainnetome atlas” (http://www.brainnetome.org/, accessed on 2 March 2020) as nodes, for a total of 123. Thus, for each participant, a 123 × 123 connectome matrix was constructed (Figure 2). Before each individual’s matrix was delivered into the graph theoretical calculation, a backbone method with a threshold of >75% was applied to minimize errors caused by normalization errors or partial volume effects [25].

### 2.8. Graph Theory Analysis

Global and nodal topological properties of contralesional hemisphere, including global efficiency (the ability of the information conduction of the entire topological network), shortest path length (average shortest length from one node to another in the network), nodal efficiency (global efficiency averaged by nodes), degree centrality (a direct measure of a single node’s centrality among the entire network), and small-world properties (representing the efficiency of global information delivery), were calculated using the GRETNA toolbox (http://www.nitrc.org/projects/gretna/, accessed on 2 March 2020) for each patient; binarized networks were used in the calculation of each topological property. The specific meaning and calculation formula for each property are provided in the Appendix A of the previous study (https://www.mdpi.com/2076-3425/12/1/60, accessed on 1 January 2022).

### 2.9. Statistical Analysis

Using GLM, two-sample t-tests were used to identify group differences in the cortical volume. Only voxels within the mask were compared with a hand-drawn mask of the contralesional hemisphere. The comparisons were as follows: (1) APs vs. mAPs; (2) APs vs. nAPs; (3) mAPs vs. nAPs; and (4) APs and mAPs vs. nAPs (voxel-wise unpaired t-tests, corrected by cluster-wise FDR, *p* < 0.05). In all group analyses, baseline characteristics, including age, gender, education level, and tumor volume, were set as covariates.

Clinical characteristics and topological properties were compared between groups using IBM SPSS statistics for Windows (v25.0, IBM^®^ Corp., Armonk, NY, USA). One-way analysis of variance (ANOVA), followed by post-hoc analysis (LSD), were used to identify significant group differences. Other analytical methods, including unpaired *t*-tests, chi-square tests, and Pearson’s correlation tests, were also used depending on the type of statistics.

Pearson’s correlation analysis and causal mediation analysis were applied to identify the factors of language performance (AQ score), using the SPSS statistics and PROCESS v3.0. (Preacher and Hayes, 2004). The parameters of causal mediation analysis were as follows: model number = 4; confidence interval = 95; number of bootstraps = 5000; mediator = the distance from the tumor to the eloquent-area; independent variable = nodal efficiency of 28/34 of Brodmann area (A28_34) and nodal efficiency of dorsomedial parietal-occipital sulcus (dmPOS); and dependent variable = AQ score.

## 3. Results

### 3.1. Demographic Characteristics

The demographic characteristics of the enrolled patients are summarized in Table 1. Sixty patients (mean age, 43.45 ± 8.47 years; sex ratio, 35 men/25 women) with a histologically proven LGG affecting the left eloquent-area were included over a period of 3 years. Based on the grouping criteria, patients were divided into AP (*n* = 15), mAP (*n* = 20), and nAP (*n* = 25) groups. The distance between the tumor border and the eloquent-area (SDTN) showed group differences between the groups, and also appeared to have a positive correlation with the AQ score (r^2^ = 0.31, *p* = 0.002). No significant intergroup differences were observed in age, sex, educational level, or tumor volume (*p* > 0.05; Table 1). The tumor location within all groups showed variations. Most tumors were centered in the frontal lobe, near the Broca area (*n* = 41), while a smaller cohort centered at the insula (*n* = 13) and the temporal lobe (*n* = 6). The tumor overlap map of each group is shown separately in Appendix A. The figure shows that the milder the patient’s aphasia symptoms, the higher the center of the tumor overlap map, and the further it is from the inferior frontal gyrus (Broca area), which is consistent with the results from the analysis on SDTN. 

### 3.2. VBM Analysis

The significant cortical clusters in the comparisons are listed in the Appendix A and are visualized in Figure 1 (rendered by xjview, https://www.alivelearn.net/xjview/, accessed on 12 February 2022). Briefly, elevated cortical volume was identified in the nAP group in two comparisons (AP vs. nAP and mAP vs. nAP). Among these comparisons, the significant clusters were located at the right SMA and right prefrontal lobe (BA 6, BA8), right gyrus rectus (BA 11), and right mesial temporal lobe (BA 20, BA 37, BA 38). Significant clusters in mAP vs. nAP were limited in the right paracentral lobule, and right SMA (BA 4, BA 5) and right gyrus rectus (BA 11), which was much less extensive than in the other two comparisons. No significant clusters were identified in the AP vs. mAP group.

### 3.3. Global Topological Properties

The results of the group comparisons of the global topological characters are summarized in Table 2 and Figure 2. Robust inter-group differences were discovered in global efficiency (one-way ANOVA, *p* = 0.003) and global shortest path length (one-way ANOVA, *p* = 0.003). The nAP had increased global efficiency compared to AP and mAP (post-hoc, LSD, *p* = 0.001 and *p* = 0.039, respectively). Meanwhile, the nAP also had a decreased path length compared with the AP and mAP (post-hoc, LSD, *p* = 0.001 and *p* = 0.049, respectively). No group differences were found in assortative, hierarchy, or synchronization. No correlations were detected between global topological properties and language scores.

### 3.4. Nodal Topological Properties

The significant results of the group comparisons of nodal topological properties are summarized in Table 3 and Figure 3. A total of 10 nodes showed inter-group differences in nodal efficiency (one-way ANOVA, FDR corrected, and *p* < 0.05). On all of these nodes, the nAP had a higher nodal efficiency than the other two groups. For A28_34_R and dmPOS_R, the nodal efficiency in mAP was also higher than that in AP. The nodal efficiency value was positively correlated with the AQ score of these nodes (r^2^ = 0.178, *p* = 0.008; r^2^ = 0.27, *p* < 0.001, respectively). All results of nodal properties are available in the Appendix A.

### 3.5. Causal Mediation Results

The results of causal mediation analysis (Table 4 and Table 5 and Appendix A) showed that the distance from the tumor to the language network was a mediating factor between the nodal efficiency of nodes A28_34 and AQ (total effect = 255.13, direct effect (DE) = 213.02, indirect effect (IE) = 42.11, DE% = 83.49%, IE% = 16.51%). Similarly, the distance from the tumor to the language network was a mediating factor between the nodal efficiency of nodes dmPOS and AQ (total effect = 495.50, DE = 439.53, IE = 55.97, DE% = 88.70%, IE% = 11.30%).

## 4. Discussion

Patients with gliomas close to eloquent area vary considerably in terms of their language deficits. The focus of this study was to identify the contralesional cortical and subcortical characteristics underlying this heterogeneity. Our results demonstrated that the non-anomia group had a higher cortical volume in the right SMA and the right mesial temporal lobe. With regard to white matter connectivity, the same group also showed the highest global efficiency and shortest global path length. Furthermore, nodes A28_34 and dmPOS had a significant correlation with the AQ score, and this association was mediated by the SDTN. We believe that our findings provide new insights into contralesional functional compensation in patients with glioma from the perspective of structural MRI.

Two hypotheses concerning our findings can be proposed. The first is activation of brain plasticity. The plasticity theory has been widely discussed in research on slow-growing lesions, mostly LGG [26,27,28]. The cellular mechanism is thought to be a creation of new synapses [29,30] and the formation of microglia cells [26], which radiologically manifested as elevated cortical volume [28,31] In this process, there should generally be two essential elements: invasion of functional regions and chronic disease course. Lesions located in functional regions, particularly in the eloquent cortex or SMAs, are more likely to trigger functional compensation than other regions [9,18,32]. These areas are heavily and efficiently activated in daily life. The repeated use of impaired function during chronic cortical damage can promote plastic activation in potential functional areas. The process is typically slow. Both animal models and imaging studies confirm that this process takes three weeks to several months; therefore, the course of chronic development is also an important factor [12,28] Another hypothesis is that some individuals may be born with contralateral compensatory potential, and their contralateral functional areas have a higher cortical volume and a stronger reserve function. Once functional damage caused by the tumor occurs, its potential functional area can quickly compensate to make up for the function of the damaged area.

The fact that slow-growing lesions lead to elevated cortical volume in the contralesional hemisphere has been reproduced multiple times [15,19,27,28], and has also been demonstrated in the current research (Figure 1). Unlike previous studies, we also observed global and local characteristics in the contralateral structural connectivity network at the white matter level and found that GM volume and WM connectivity were consistent. At the whole-brain level, compared with the nAP group, the cortical volume of multiple brain regions in the contralesional hemisphere and the global efficiency of white matter connectivity increased. An increase in cortical volume represents an increase in cortical function, while an increase in white matter network efficiency and a decrease in shortest path length imply the formation of long-range connections [33]. At the local level, the cortex volume of the medial temporal lobe and parieto-occipital junction increased significantly in the AP group compared with the nAP group, and the nodal efficiency of these regions also increased significantly. These results demonstrate that both cortical volume and WM network efficiency are related to the degree of aphasia, which is related to functional compensation, as previously stated.

Naming ability is a combination of language, motor, visual, and memory functions. In contrast to previous findings, our results showed that, except for the homotopic areas, some other cortices (orbitofrontal gyrus, SMA, and medial temporal gyrus) in the contralesional hemisphere had increased volumes. These cortices were commonly found to increase in volume in stroke or glioma patients with recovery of naming ability. (i) The orbitofrontal gyrus is involved in decision-making, higher emotion, and execution [34,35]. Some studies have also reported that it has complex structural and functional connections with the language functional area [36]. (ii) SMA is responsible for integrating and controlling motor functions and plays a crucial role in functional compensation [32]. (iii) Moreover, the mesial temporal lobe, pole of the temporal lobe, and hippocampus are responsible for memory and retrieval. Since we only observed restricted cortical volume increases in the mAP group, we hypothesized that the contralesional SMA and orbitofrontal lobe may be involved in the compensation of naming function. The extensive cortical differences found in the comparison of the AP with the nAP may be related to the compensation of other complex language functions.

Similar results were observed for white matter connectivity networks. The nodal efficiency of ten nodes was significantly higher in the nAP group than in the other two groups, indicating that in the nAp group, the information processing efficiency of these cortical nodes was increased, which is also mainly due to the establishment of effective fiber connections of nodes to other nodes. Moreover, the spatial positions of the ten nodes overlapped with a large part of the cortical thickening area, which means that the compensation of the GM and WM is consistent. This is the rise in connections between these nodes, which leads to an overall change in global network efficiency.

The SDTN was evaluated by an awake craniotomy approach, which is currently the most accurate way to measure this index. The results of analysis on SDTN and the tumor overlap maps of each group (Appendix A) both indicated that the shorter the distance, the higher chance of language deficit caused by the glioma, which is consistent with previous studies [8,37]. Thus, SDTN is one of the major factors affecting the degree of preoperative aphasia, so we did not set this value as a covariate in analytic models. Our results showed that the altered nodal efficiency of contralateral cortices located in the para-hippocampus and parietal-occipital sulcus affected language performance partially through the distance from glioma to language network. The shorter the distance, the lower the nodal efficiency of these two nodes and the lower the language performance. As described above, the cortex located at the para-hippocampus and parietal-occipital sulcus are responsible for memory and visual functions related to naming ability, respectively. These findings imply that the glioma, which is near the language network, caused the nodal efficiency of these two sites to be low, and these two sites were unable to participate in language compensation. Other clinical or radiological features, including age, education level and tumor size, showed neither inter-group differences nor significant correlation with AQ value, so those data were considered as covariates in subsequent analysis.

The findings of this study provide evidence for compensation of contralateral language function in gliomas. However, there are some limitations. We propose two hypotheses for contralateral functional compensation in the discussion; however, due to the lack of longitudinal data, we cannot determine which hypothesis dominates the results of this study. Further studies are needed to incorporate longitudinal, multi-time-point data to better explain the specific mechanisms of contralateral functional compensation.

## 5. Conclusions

The present study observed an increase in cortical volume and subcortical network efficiency in eloquent-area glioma patients; the cortex with increased volume included the contralesional SMA, mesial temporal lobe, and orbitofrontal lobe. The nodes with increased nodal efficiency showed a special correlation with an increased AQ value, and the process could possibly be mediated by the increased SDTN. These results may be associated with functional compensation.

## Figures and Tables

**Figure 1 cancers-14-04469-f001:**
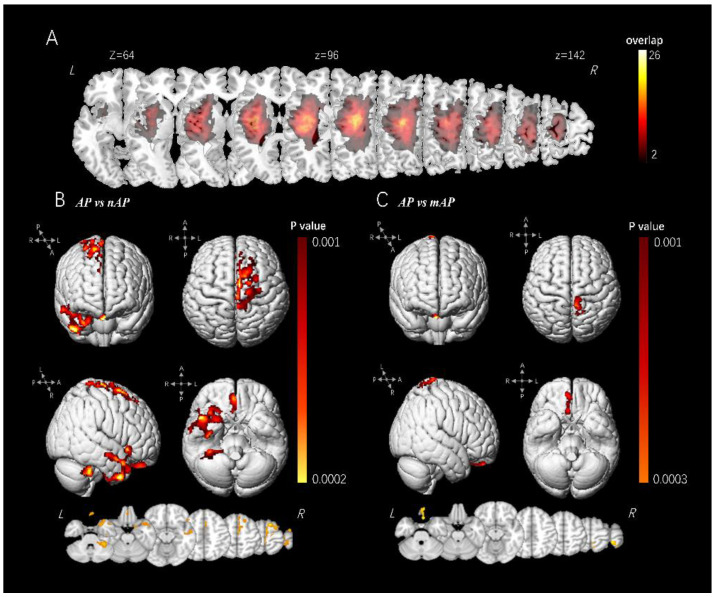
Tumor overlapping map and regions with group difference. (**A**): The tumor overlapping map of all enrolled patients. The color of each voxel indicates the number of overlap (1–26). (**B**,**C**): Significant clusters of inter-group comparisons identified by VBM, with 3-D rendered figures on the top and transverse map on the bottom. The color indicates *p* value, ranging from 0.001 (dark red) to 0.0002 (light yellow).

**Figure 2 cancers-14-04469-f002:**
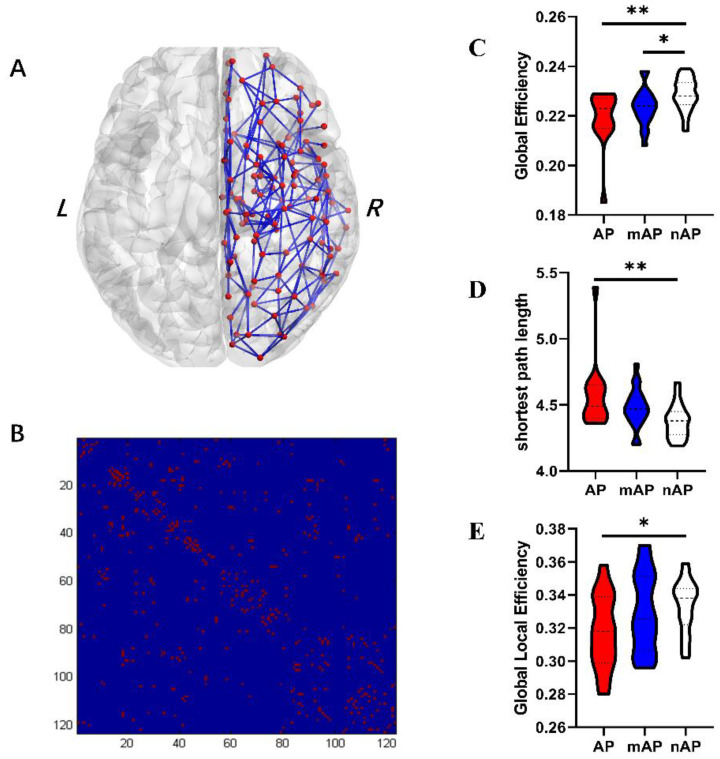
(**A**): All contralesional nodes and white matter connections identified by deterministic fiber tracking with the backbone method. (**B**): Binarized connectome matrix of the contralesional hemisphere. (**C**–**E**): Violin figures of global topological properties with group differences. (*, *p* < 0.05; **, *p* < 0.01).

**Figure 3 cancers-14-04469-f003:**
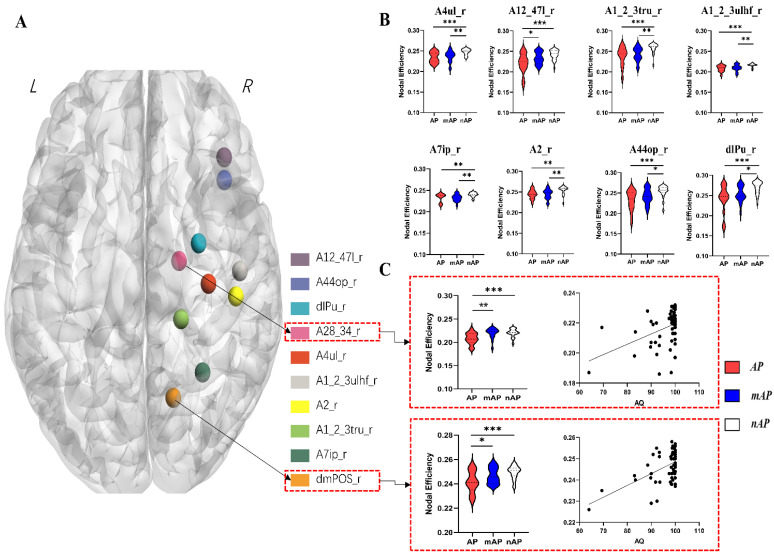
(**A**): A 3-D rendered figure showing all ten nodes with inter-group differences. (**B**): Violin figures of eight nodes that showed significant inter-group differences, the nodal efficiency in the nAP group is higher than AP group at all nodes. (**C**): The nodal efficiency in two nodes (A28_34 and dmPOS) showed not only group differences but also positive correlation with AQ value. (*, *p* < 0.05; **, *p* < 0.01; and ***, *p* < 0.001).

**Table 1 cancers-14-04469-t001:** Demographic Characteristics.

Group	AP	mAP	nAP	*p* Value
Sex				0.39
Male	7	11	17	
Female	8	9	8	
Age (yrs.)	47.8 ± 11.5	43.9 ±11.0	41.5 ± 7.3	0.15
Education Level (yrs.)	11.7 ± 2.9	12.7 ± 2.9	13.6 ± 2.9	0.12
Pathology				0.65
Astrocytoma	8	13	13	
Oligodendroglioma	7	7	12	
Tumor Volume (CC)	33.34 ± 23.91	31.74 ± 19.90	26.22 ± 19.38	0.51
SDTN (mm)	5.66 ± 1.39	11.92 ± 6.00	15.40 ± 8.86	<0.01
AQ Score	8.65 ± 0.48	9.36 ± 1.67	10.00 ± 0.00	<0.01
Naming Score	87.05 ± 8.92	98.68 ± 0.52	99.91 ± 0.25	<0.01

AP, aphasia group; mAP, mild aphasia group; nAP, normal group; SDTN, shortest distance from tumor to language network; and AQ, aphasia quotient.

**Table 2 cancers-14-04469-t002:** Summary of Global topological properties.

Property	Value (Mean ± Standard Deviation)			One-Way ANOVA (*p* Value)	Post-Hoc Analysis with LSD (*p* Value)		
	AP	mAP	nAP		AP vs. mAP	AP vs. nAP	mAP vs. nAP
Global efficiency	0.219 ± 0.011	0.224 ± 0.007	0.225 ± 0.009	0.003	0.124	0.001	0.039
Global local efficiency	0.318 ± 0.022	0.329 ± 0.237	0.333 ± 0.016	0.075	0.110	0.025	0.508
Shortest path length	4.570 ± 0.254	4.479 ± 0.148	4.374 ± 0.125	0.003	0.123	0.001	0.049
Assortativity	1.601 ± 0.716	1.449 ± 0.520	1.469 ± 0.596	0.715	0.448	0.488	0.913
Hierarchy	3.587 ± 0.686	3.644 ± 0.408	3.779 ± 0.576	0.532	0.767	0.424	0.077
Synchronization	0.034 ± 0.912	0.176 ± 1.151	0.328 ± 0.920	0.246	0.540	0.323	0.099

AP, aphasia group; mAP, mild aphasia group; nAP, normal group; LSD, least significant difference; and ANOVA, analysis of variance.

**Table 3 cancers-14-04469-t003:** Significant alterations of nodal efficiency.

Node	Value (Mean ± Standard Deviation)			One-Way ANOVA (*p* Value)	Post-Hoc Analysis with LSD (*p* Value)		
	AP	mAP	nAP		AP vs. mAP	AP vs. nAP	mAP vs. nAP
A12_47l_r	0.221 ± 0.024	0.233 ± 0.015	0.241 ± 0.012	0.002	0.041	<0.001	0.098
A1_2_3tru_r	0.238 ± 0.023	0.243 ± 0.017	0.258 ± 0.012	<0.001	0.445	<0.001	0.003
A1_2_3ulhf_r	0.208 ± 0.008	0.209 ± 0.008	0.216 ± 0.004	<0.001	0.580	0.001	0.002
A28_34_r	0.208 ± 0.012	0.220 ± 0.002	0.220 ± 0.008	0.002	0.002	0.001	0.856
A2_r	0.244 ± 0.011	0.245 ± 0.012	0.254 ± 0.010	0.003	0.865	0.004	0.003
A44op_r	0.232 ± 0.024	0.241 ± 0.019	0.255 ± 0.013	0.001	0.176	<0.001	0.017
A4ul_r	0.234 ± 0.013	0.237 ± 0.014	0.248 ± 0.007	<0.001	0.482	0.001	0.002
A7ip_r	0.233 ± 0.010	0.233 ± 0.008	0.240 ± 0.005	0.003	0.893	0.006	0.002
dlPu_r	0.242 ± 0.033	0.253 ± 0.020	0.270 ± 0.015	<0.001	0.134	<0.001	0.017
dmPOS_r	0.242 ± 0.009	0.247 ± 0.007	0.249 ± 0.005	0.003	0.030	0.001	0.198

A12_47l_r, lateral BA 12/47; A1_2_3tru_r, BA 1/2/3 trunk region; A1_2_3ulhf_r, BA 1/2/3 upper limb, head and face region; A28_34_r, BA 28/34; A2_r, BA 2; A44op_r, opercula BA 44; A4ul_r, BA 4 upper limb region; A7ip_r, intraparietal BA 7; dlPu_r, dorsolateral putamen; dmPOS_r, dorsomedial parietal-occipital sulcus; AP, aphasia group; mAP, mild aphasia group; nAP, normal group; LSD, least significant difference; and ANOVA, analysis of variance.

**Table 4 cancers-14-04469-t004:** Testing the mediation effect of nodal efficiency of A28_34 in the healthy hemisphere on language performance.

	Effect	SE or Boot SE	*t* Value	*p* Value	Lower Limited 95% CI	Upper Limited 95% CI	Percentage of Effect
Distance to language network							
Constant	−28.68	18.43	−1.56	0.1252	−65.58	8.22	-
Nodal efficiency A28_38	181.47	84.83	2.14	0.0366	11.66	351.28	-
Indirect effect model							
Constant	47.57	15.78	3.02	0.0038	15.97	79.16	-
Distance	0.23	0.11	2.11	0.0395	0.01	0.45	-
Nodal efficiency A28_38	213.02	73.88	2.88	0.0055	65.07	360.97	-
Total effect model							
Constant	40.91	15.90	2.57	0.0127	9.07	72.76	-
Nodal efficiency A28_38	255.13	73.21	3.48	0.0009	108.58	401.68	-
Summary							
Total effect	255.13	73.21	3.48	0.0009	108.58	401.68	-
Direct effect	213.02	73.88	2.88	0.0055	65.07	360.97	83.49%
Indirect effect	42.11	18.90	-	<0.05	9.06	82.93	16.51%

A28_38 = 28/34 (EC, entorhinal cortex) of Brodmann area, CI = confident interval.

**Table 5 cancers-14-04469-t005:** Testing the mediation effect of nodal efficiency of dmPOS in the healthy hemisphere on language performance.

	Effect	SE or Boot SE	*t* Value	*p* Value	Lower Limited 95% CI	Upper Limited 95% CI	Percentage of Effect
Distance to language network							
Constant	−56.31	32.24	−1.75	0.0860	−120.85	8.23	-
Nodal efficiency dmPOS	272.01	130.81	2.08	0.0420	10.16	533.87	-
Indirect effect model							
Constant	−14.21	26.01	−0.55	0.5870	−66.28	37.87	-
Distance	0.21	0.10	1.99	0.0499	0.01	0.41	-
Nodal efficiency dmPOS	439.53	106.61	4.12	0.0001	226.05	653.01	-
Total effect model							
Constant	−25.79	25.99	−0.99	0.3251	−77.82	26.23	-
Nodal efficiency dmPOS	495.50	105.45	4.70	<0.0001	284.43	706.57	-
Summary							
Total effect	495.50	105.45	4.70	<0.0001	284.43	706.57	-
Direct effect	439.53	106.61	4.12	0.0001	226.05	653.01	88.70%
Indirect effect	55.97	19.90	-	<0.05	16.50	94.87	11.30%

CI = confident interval, dmPOS = dorsomedial parietal occipital sulcus.

## Data Availability

Anonymized data and material will be available on reasonable request.

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
