# Peer review of "Contralesional Cortical and Network Features Associated with Preoperative Language Deficit in Glioma Patients"

_cancers, 2022, doi:10.3390/cancers14184469_

Round 1

Reviewer 1 Report

This study assesses the effect that low grade gliomas in the dominant hemisphere, close to language areas, have on the contralateral white matter connectivity and grey matter.

The study is well designed and is an interesting addition to the body of evidence on plasticity caused by low grade gliomas.

I think the work would improve with some minor change.

1- The authors mention tumors anchored in eloquent areas without saying where they are centered. From Fig1 it looks like they are mostly centred in the left insula, with large variations since out of 60 patients only a maximum of 26 has overlapping tumor areas. The authors should present the tumor overlapping maps for the subgroups AP, nAP, mAP to allow for an assessment of possible differences. They should also mention that the tumors are mostly centred in the left insula (if this is the case) since the most common language eloquent areas one thinks of are Broca (inferior frontal gyrus) and Wernicke (angular/supramarginal gyrus) rather than insula.

2- It is known that the orbito-frontal region is connected to the temporal pole via the Uncinate Fasciculus (UF), which runs through the insula. Is there a difference between AP, mAP and nAP in how luckily it is that the tumour has damaged the UF in the left hemisphere? The authors have all the data to be able to dissect this tract from the diffusion data they have acquired. another fascicle running through the insula is the inferior fronto-occipital fasciculus (IFOF). Even if the authors don't want to segment these tracts, there are atlases available (e.g. the Julich histological atlas, also available on FSL and SPM) at least of the UF. Are some of the clusters identified by VBM connected by the UF? 

There are some minor typos especially in the simple summary where gliomas are said to "exhibit" deficits, whilst gliomas cause deficits, are patients who exhibit them. The rest of the paper is well written.

I feel that in the methods it should be mentioned that VBM and connectivity are assessed in the contralateral hemisphere, I only realised this towards the results session and I was puzzled when reading the methods regarding how the tumor was assessed with these techniques. 

More comments are in the attached file.

Author Response

This study assesses the effect that low grade gliomas in the dominant hemisphere, close to language areas, have on the contralateral white matter connectivity and grey matter.

The study is well designed and is an interesting addition to the body of evidence on plasticity caused by low grade gliomas.

I think the work would improve with some minor change.

1- The authors mention tumors anchored in eloquent areas without saying where they are centered. From Fig1 it looks like they are mostly centred in the left insula, with large variations since out of 60 patients only a maximum of 26 has overlapping tumor areas. The authors should present the tumor overlapping maps for the subgroups AP, nAP, mAP to allow for an assessment of possible differences. They should also mention that the tumors are mostly centred in the left insula (if this is the case) since the most common language eloquent areas one thinks of are Broca (inferior frontal gyrus) and Wernicke (angular/supramarginal gyrus) rather than insula.

Thanks for your comment. Most of the tumor centered at the prefrontal lobe, close to Broca area. The tumors could press down on the insula, and forcing the insula lobe to move to deeper location. When we overlap the tumor masks on the standard brain template, it may look like large amounts of tumor centered at the insula. In fact, there were only 13 cases within all three groups that centered at the insula (AP n=4, mAP n=6, nAP n=3). As suggested, we presented the overlapping map for all three subgroups below. The figure shows that the milder the patient's aphasia symptoms, the higher the center of the tumor overlap map, and the further it is from the inferior frontal gyrus, which includes the Broca area. So, as a result, the figure is consistent with the results of SDTN measurements. We have added the description in results (page 5, line 198-204) and discussion (page 11, line 337-340). The figure is added in supplementary meterial.

2- It is known that the orbito-frontal region is connected to the temporal pole via the Uncinate Fasciculus (UF), which runs through the insula. Is there a difference between AP, mAP and nAP in how luckily it is that the tumour has damaged the UF in the left hemisphere? The authors have all the data to be able to dissect this tract from the diffusion data they have acquired. another fascicle running through the insula is the inferior fronto-occipital fasciculus (IFOF). Even if the authors don't want to segment these tracts, there are atlases available (e.g. the Julich histological atlas, also available on FSL and SPM) at least of the UF. Are some of the clusters identified by VBM connected by the UF? 

Thanks for the advice. As suggested, we segmented the IFOF and UF according to JHU-white matter atlas (ipisilateral to tumor), and extracted the mean diffusion metrics of these tracts (Fractional Anisotropy and Mean Diffusivity). One-way ANOVA was used to identify inter-group differences. The figures and tables showing our results are listed below. No significant inter group differences at these tracts was found. But interestingly, from the figure we noticed that some individuals (around 2-4 with in each group) have extra-ordinary high MD and low FA, which strongly indicated robust damage of the corresponding fiber tracts. We believe those may be the tumor-damaged fiber tracts, as the reviewer indicated. (We have added the table and figure into supplementary material but not in the manuscript, because it has little to do with the theme of the current research.)

 Figure S2: Mean Diffusion metrics of IFOF and UF. Statistical analysis showed no inter-group differences. But the figure showed some individuals (around 2-4 with in each group) have extra-ordinary higher MD and lower FA, which strongly indicated robust damage of the corresponding fiber tracts.

Table S5: Mean Diffusion metrics of IFOF and UF.

Diffusion Metrics

Value (Mean ± Standard deviation)

One-way ANOVA

(p value)

AP

mAP

nAP

Mean FA of UF

0.385 ± 0.036

0.372 ± 0.062

0.390 ± 0.035

0.403

Mean MD of UF (*103)

0.819 ± 0.093

0.858 ± 0.131

0.828 ± 0.109

0.552

Mean FA of IFOF

0.409 ± 0.042

0.381 ± 0.064

0.406 ± 0.031

0.127

Mean MD of IFOF (*103)

0.809 ± 0.077

0.846 ± 0.147

0.825 ± 0.091

0.597

* FA, Fractional Anisotropy; MD, Mean Diffusivity; UF, Uncinate Fasciculus; IFOF, Inferior Frontal Occipital Fasciculus; AP, aphasia group; mAP, mild aphasia group; nAP, normal group; ANOVA, analysis of variance.

For the last question, due to tumor mass effect, it is difficult to perform group analysis of glioma-invaded region and its surrounding cortex, so we only performed VBM on the contralesional hemisphere. As shown in Figure1 and supplementary Table S4, the significant clusters located at BA 38 (Temporal pole) and BA 11 (orbital frontal lobe) is possibly connected by contralesional UF.

There are some minor typos especially in the simple summary where gliomas are said to "exhibit" deficits, whilst gliomas cause deficits, are patients who exhibit them. The rest of the paper is well written.

Thanks for pointing out the errors. We have revised those accordingly.

I feel that in the methods it should be mentioned that VBM and connectivity are assessed in the contralateral hemisphere, I only realised this towards the results session and I was puzzled when reading the methods regarding how the tumor was assessed with these techniques. 

Thank you for pointing out, we have added the corresponding description in the method section. (page 3, line 135-137; page 4, line 147; page 4, line 152-154.)

More comments are in the attached file.

All comments were read by the authors and we revised the paper accordingly.

Reviewer 2 Report

In this paper Increasing contralesional hemispheric cortical volume and net-2 work efficiency of the subcortical network contribute to avoid 3 preoperative language deficit, the authors describe methods to determine potential compensations in the contralateral cerebral hemisphere in patients who have language deficits due to infiltration by gliomas.

Overall, this is an interesting paper that uses a total of 60 patients with grade II and III gliomas with involvement of language areas in the dominant brain hemisphere to show that patients with anomia, mild anomia, and normal naming function had differences in the contralateral brain hemisphere for brain volume and DTI based connectivity and efficiency measures.

There are a few significant points that should be addressed:

1.       There is some lack of clarity about sidedness throughout the paper, and it should be very clear in the figures and text when they are referring to ipsilesional or contralesional differences. This creates a problem with the figures, which are the opposite of radiology standard. If they are going to be shown this way, they should be labeled with ‘R’ and ‘L’ and probably should anyway.

2.       Much of the differences between the groups could potentially be explained by tumor size and shortest distance between the tumor and the eloquent area. Did the authors attempt to adjust for this in their model? This should be addressed in the discussion either way.

Overall though I believe this is an instructive and interesting paper which will be a valuable contribution to the literature.

Please see more specific comments listed below:

3.       The title is somewhat speculative that the mechanism of the differences detected is due to avoiding a language deficit. It would be more straightforward if it simply stated those features are associated with less preoperative deficit.

Abstract:

4.       Abstract says diffusion sequences when they may mean “diffusion tensor imaging” more accurately.

5.       If the paper is addressing lower grade gliomas (grade II or III only), it should be put in the abstract.

Introduction:

6.       Ok as written.

Methods:

7.       Eloquent and non-eloquent areas (line 80) should be more specifically defined. If an atlas was used, which areas were noted as eloquent?

8.       Was the surgical technique used only to determine the shortest distance to the tumor? Also, I’m not sure what the word “engraving” (line 106) really means. Please clarify.

Results:

9.       The sidedness of relevant findings should be noted in section 3.2 (lines 197-204).

Discussion:

10.   Sidedness of mesial temporal lobe (line 261 should be stated).

11.   As above, please put some description of whether the authors attempted to adjust for size and SDTN in their model.

Figures/Tables:

12.   Figure 1. It is not clear from this figure which brain regions are right and left. The top image appears to be the opposite of normal radiology notation, which is to put the patient left on the image right. Similarly, I’m not sure of the directionality in B-C.

13.   Directionality should be explicitly labeled in the remaining figures as well.

Author Response

In this paper Increasing contralesional hemispheric cortical volume and network efficiency of the subcortical network contribute to avoid 3 preoperative language deficit, the authors describe methods to determine potential compensations in the contralateral cerebral hemisphere in patients who have language deficits due to infiltration by gliomas.

Overall, this is an interesting paper that uses a total of 60 patients with grade II and III gliomas with involvement of language areas in the dominant brain hemisphere to show that patients with anomia, mild anomia, and normal naming function had differences in the contralateral brain hemisphere for brain volume and DTI based connectivity and efficiency measures.

There are a few significant points that should be addressed:

  1. There is some lack of clarity about sidedness throughout the paper, and it should be very clear in the figures and text when they are referring to ipsilesional or contralesional differences. This creates a problem with the figures, which are the opposite of radiology standard. If they are going to be shown this way, they should be labeled with ‘R’ and ‘L’ and probably should anyway.

Thank you for pointing out this problem, we have now re-labeled all the figures with L and R.

  1. Much of the differences between the groups could potentially be explained by tumor size and shortest distance between the tumor and the eloquent area. Did the authors attempt to adjust for this in their model? This should be addressed in the discussion either way.

Thanks for your comment, the tumor volume showed neither inter-group difference nor correlation with AQ value, and we set it as a covariate in the GLM in further group analysis, as we have stated this point in original manuscript. The distance showed relationship with both AQ value and nodal topological properties, and has significantly inter-group difference. We assume that this may be one of the major factor that influenced functional compensation, so we didn’t set this data as a covariate. we addressed this point in the discussion as suggested. (page 11, line337-343 and page 11 line 351-354)

Overall though I believe this is an instructive and interesting paper which will be a valuable contribution to the literature.

Please see more specific comments listed below:

  1. The title is somewhat speculative that the mechanism of the differences detected is due to avoiding a language deficit. It would be more straightforward if it simply stated those features are associated with less preoperative deficit.

Thanks for your advice! The title have now changed into “Contralesional cortical and network features associated with preoperative language deficit in glioma patients.”

Abstract:

  1. Abstract says diffusion sequences when they may mean “diffusion tensor imaging” more accurately.

We revised the expression as the reviewer suggested.(page 1, line 28)

  1. If the paper is addressing lower grade gliomas (grade II or III only), it should be put in the abstract.

We added the content as the reviewer suggested.(page 1, line 23)

Introduction:

  1. Ok as written.

Methods:

  1. Eloquent and non-eloquent areas (line 80) should be more specifically defined. If an atlas was used, which areas were noted as eloquent?

Thanks for your advice, the pre-surgical radiological assessment was based Broadmann atlas, Broca and Wernicke area was defined as eloquent area (BA 39,40,44,45). We have added the corresponding description in methods section. (page 2, line 82-83)

  1. Was the surgical technique used only to determine the shortest distance to the tumor? Also, I’m not sure what the word “engraving” (line 106) really means. Please clarify.

The main purpose of awake craniotomy was to maximize the extent of tumor resection while minimize the damage of eloquent area, the SDTN was only “by-products” of this procedure.  We used the intra-operative SDTN data for analysis because it is more accurate than any radiological measurements. The word “engraving” may cause confusion, we just wanted to say that veins on the brain surface were all carefully preserved, thus, we revised the manuscript accordingly. (page 3, line 107-109)

Results:

  1. The sidedness of relevant findings should be noted in section 3.2 (lines 197-204).

The sidedness is noted according to the reviewer’s suggestion.(page 5, line 213-217)

Discussion:

  1. Sidedness of mesial temporal lobe (line 261 should be stated).

The sidedness is noted according to the reviewer’s suggestion. (page 10, line 276)

  1. As above, please put some description of whether the authors attempted to adjust for size and SDTN in their model.

Figures/Tables:

  1. Figure 1. It is not clear from this figure which brain regions are right and left. The top image appears to be the opposite of normal radiology notation, which is to put the patient left on the image right. Similarly, I’m not sure of the directionality in B-C.

The sidedness and directionality were all noted according to the reviewer’s suggestion.

  1. Directionality should be explicitly labeled in the remaining figures as well.

The sidedness and directionality were all noted according to the reviewer’s suggestion.
